# Exercise Affects Blood Glucose Levels and Tissue Chromium Distribution in High-Fat Diet-Fed C57BL6 Mice

**DOI:** 10.3390/molecules25071658

**Published:** 2020-04-03

**Authors:** Geng-Ruei Chang, Po-Hsun Hou, Wen-Kai Chen, Chien-Teng Lin, Hsiao-Pei Tsai, Frank Chiahung Mao

**Affiliations:** 1Department of Veterinary Medicine, National Chiayi University, 580 Xinmin Road, Chiayi 60054, Taiwan; grchang@mail.ncyu.edu.tw (G.-R.C.); vet540423@gmail.com (C.-T.L.); 2Veterinary Teaching Hospital, National Chiayi University, 580 Xinmin Road, Chiayi 60054, Taiwan; tsaibelle@mail.ncyu.edu.tw; 3Department of Psychiatry, Taichung Veterans General Hospital, 4 Section, 1650 Taiwan Boulevard, Taichung 40705, Taiwan; peterhopo2@yahoo.com.tw; 4Faculty of Medicine, National Yang-Ming University, 2 Section, 155 Linong Street, Beitou District, Taipei 11221, Taiwan; 5Department of Veterinary Medicine, National Chung Hsing University, 250 Kuo Kuang Road, Taichung 40227, Taiwan; 210902@mail.ttl.com.tw

**Keywords:** high-fat diet, chromium, exercise, blood glucose, obesity

## Abstract

Obesity is commonly associated with hyperglycemia and type 2 diabetes and negatively affects chromium accumulation in tissues. Exercise prevents and controls obesity and type 2 diabetes. However, little information is available regarding chromium changes for regulating glucose homeostasis in high-fat diet (HFD)-fed animals/humans who exercise. Therefore, this study explored the effects of exercise and whether it alters chromium distribution in obese mice. Male C57BL6/J mice aged 4 weeks were randomly divided into two groups and fed either an HFD or standard diet (SD). Each group was subgrouped into two additional groups in which one subgroup was exposed to treadmill exercise for 12 weeks and the other comprised control mice. HFD-fed mice that exercised exhibited significant lower body weight gain, food/energy intake, daily food efficiency, and serum leptin and insulin levels than did HFD-fed control mice. Moreover, exercise reduced fasting glucose and enhanced insulin sensitivity and pancreatic β-cell function, as determined by homeostasis model assessment (HOMA)-insulin resistance and HOMA-β indices, respectively. Exercise also resulted in markedly higher chromium levels within the muscle, liver, fat tissues, and kidney but lower chromium levels in the bone and bloodstream in obese mice than in control mice. However, these changes were not noteworthy in SD-fed mice that exercised. Thus, exercise prevents and controls HFD-induced obesity and may modulate chromium distribution in insulin target tissues.

## 1. Introduction

The growing prevalence of obesity and its association with diabetes, hypertension, hyperlipidemia, and cardiovascular disease has made it a major public health epidemic [1,2,3]. Energy components establish a balance between energy intake and expenditure [4,5]. When energy intake equals energy output, body weight remains constant. However, an imbalance in one direction results in a net increase in body energy storage, which can lead to body weight gain and possibly obesity [5]. Treating and preventing obesity require reducing food intake and/or enhancing and increasing energy expenditure or output. Physical activity is the most variable and easily altered component of total energy expenditure. Therefore, increasing physical activity, such as through exercise, is often prescribed to individuals seeking to control and lose body weight. By contrast, physical inactivity is considered a potential risk factor for type 2 diabetes and other metabolic diseases, whereas exercise (particularly endurance exercise training) may enhance fat oxidation, which is associated with insulin sensitivity improvement in patients with obesity [6]. However, using humans as research models to determine whether weight control is influenced by exercise alone or other factors such as diet; genetics; environmental interaction; and biological, psychological, and sociological factors is difficult.

Chromium is a key trace element that is used in various cellular functions involving carbohydrates, lipids, and protein metabolism [7]. Studies have reported that chromium exerts significantly beneficial effects on the insulin system and appears to increase insulin signaling and its proposed mechanisms [8]. In recorded clinical data, administering chromium compounds improved insulin sensitivity in patients with type 2 diabetes [4]. Moreover, chromium supplementation reduced serum glucose levels after modified forced swimming tests for diabetic rats [9]. In addition, a combined chromium and conjugated linoleic acid supplement did not augment glycogen synthesis during recovery from high-intensity exercise or after high-carbohydrate consumption in young overweight women [10]. Nevertheless, chromium supplementation did not significantly affect plasma glucose and serum insulin levels in moderately obese women who were placed on an exercise regimen [11]. Otherwise, experimental data from healthy individuals explicitly indicated a lack of effect of chromium on sugar and lipid metabolism [12].

Given the aforementioned points, exercise is generally considered to positively influence the association between glucose and chromium and is subsequently applied toward preventing or treating obesity and diabetes when using chromium supplements. The interaction between exercise and tissue chromium should be investigated but not indirectly through chromium agent supplementation. The extent of chromium use has not been fully explored regarding changes in obese individuals and other individuals after exercise training, particularly when paralleled with high glucose levels. Animal model studies can offer tighter controls than human studies by reducing some confounding factors that may complicate the interpretation of human studies [13]. Thus, we used an animal model to determine whether the response to exercise contributes to the metabolic effects of chromium changes in mice. Furthermore, we developed an obese animal model by using mice fed with a high-fat diet (HFD). The effects of endurance exercise training on HFD-induced obese animals were evaluated for metabolic parameters, blood glucose, insulin profile, and the distribution and movement of chromium.

## 2. Results

### 2.1. Metabolic Effects of Exercise on Body Weight, Food/Energy Intake, and Leptin

In our previous findings, HFD impaired glucose tolerance, induced the redistribution of metabolic tissue chromium levels, and changed the glucose metabolism [4,7,14]. In the present study, mice that underwent exercise training on an HFD for 12 weeks exhibited significant differences in body weight, weekly body weight gain, daily food/energy intake, and serum leptin levels by the end of the experiment when compared with the control group. Moreover, the body weight of exercise-treated mice was significantly lower by the end of the treatment than that of control mice (27.30 ± 0.56 versus 30.04 ± 0.76 g, respectively, *p* < 0.05) on an HFD for 12 weeks (Figure 1A). Weekly body weight gain after 12 weeks in HFD-fed mice undergoing exercise was 22% lower than that of HFD control mice, which was significantly different (0.85 ± 0.03 versus 1.09 ± 0.07 g/week, respectively, *p* < 0.01; Figure 1B). In addition, HFD-fed mice that underwent exercise presented a significantly decreased mean daily food intake (3.51 ± 0.09 versus 3.92 ± 0.08 g/mouse/d, respectively, *p* < 0.05; Figure 1C). This result paralleled the decreased daily caloric intake of the HFD group that exercised (15.80 ± 0.42 versus 17.63 ± 0.35 kcal/mouse/d, respectively, *p* < 0.05) when compared with the corresponding values of the HFD group (Figure 1D). In addition, exercise-trained mice that were fed an HFD reduced their daily food efficiency (0.0346 ± 0.0017 versus 0.0399 ± 0.0021 g bw/g food, *p* < 0.05; Figure 1E). Exercise-trained mice on an HFD exhibited a significant 3.3-fold decrease (*p* < 0.01) in the serum leptin level compared with HFD-fed mice (Figure 1F). Consistent with this result, we observed lower body weight gain in HFD-fed mice that underwent exercise than in HFD-fed mice that did not exercise, which could be attributed to decreased food or calorie intake.

### 2.2. Exercise Affected Blood Glucose and Insulin Concentration

To examine the effect of exercise on glucose homeostasis in mice fed an HFD, fasting glucose and serum insulin levels were measured after 12 weeks of exercise training. Exercise-trained mice on an HFD showed significantly lower blood glucose levels after 12 weeks of treadmill training (*p* < 0.05) and decreased blood glucose levels by approximately 17% (Figure 2A) than did obese mice fed an HFD. Moreover, the serum insulin level in obese mice was significantly different (*p* < 0.05), and exercise-trained mice had a 24% lower value than did control mice fed an HFD (Figure 2B). These results suggested that exercise can have reverse diet-induced hyperinsulinemia at up to 16 weeks of age.

### 2.3. Exercise Affected Insulin Sensitivity and β-Cell Function Indices

The aforementioned data indicated that exercise exerted beneficial effects on the glucose level in obese mice. Therefore, we investigated insulin resistance and pancreatic islet β-cell secretion function indices through a homeostasis model assessment (HOMA)-insulin resistance (IR) and HOMA-β indices, respectively, after obese mice underwent exercise. Mice on an HFD after exercise displayed positive effects that made their insulin resistance significantly lower (*p* < 0.05) during HOMA-IR by approximately 36% than did control mice (Figure 3A). Furthermore, the HOMA-β index for obese mice that underwent exercise was significantly different (*p* < 0.05). In addition, exercise-trained mice had 31% higher values than did control mice fed an HFD (Figure 3B).

### 2.4. Exercise Affected Chromium Distribution in the Tissue

To evaluate whether mice that exercised and showed alternating glucose levels could respond to changes in the chromium concentration, we assessed whether chromium level differences were related to exercise alterations in obese mice (Table 1). After 12 weeks of experimental training, exercise treatment caused significant differences in the chromium level of the tissue between the control and exercise-trained groups. The chromium level in blood and bones was significantly lower in exercise-trained mice than in control mice by approximately 12% and 22% (*p* < 0.05), respectively. This was in contrast with the chromium level in mice on an HFD that underwent exercise. The results revealed that the chromium level significantly increased by 17% in the muscle, 62% in the liver, 44% in the epididymal fat tissue, and 30% in the kidney (*p* < 0.05) over the respective chromium levels of the control group. Thus, we determined that exercise dramatically changed the chromium metabolism with a net movement of chromium in obese mice fed an HFD.

## 3. Discussion

This study determined the effects of exercise training on mice fed an HFD, particularly on mice undergoing long-term consumption of an HFD that induces obesity. Moreover, our previous findings indicated that mice fed an HFD exhibited higher glucose intolerance and a lower chromium concentration in the liver, muscle, fat, and bones than did mice fed the standard diet (SD) [4,14]. The results revealed that exercise markedly reduced the body weight and food intake of mice fed an HFD compared with HFD-fed control mice that did not exercise. Subsequently, we used the HFD-induced obese animal model that underwent exercise to reduce blood glucose levels, serum insulin concentration, and HOMA-IR indices. The HOMA-β indexes of obese mice after exercise were higher than those of control mice. Moreover, when the chromium concentration in the blood and bones was compared between the exercise-treated and control groups, the two analytic levels were lower in the exercise group. In particular, exercise-trained obese mice exhibited better muscle, liver, fat pads, and kidneys than did HFD-fed control mice.

In line with previous findings that indicated that increasing energy intake and/or reducing energy expenditure led to becoming overweight and even obese [4,5,15], physical activities were performed for physical fitness, including physical work and any muscular activity [16]. Here, exercise may have effectively prevented or reduced body weight gains in mice on an HFD for 12 weeks compared with HFD control mice. The result that body weight losses may have been due to increased energy expenditure is similar to the result reported by Møller et al. [17]. They reported that exercise is effective in increasing energy expenditure or output and helps reduce fat accumulation. Moreover, additional reports have indicated that exercise can enhance the metabolic rate, including the basal metabolic rate and resting metabolic rate, which promote the loss of fat mass and body weight [18]. Thus, the ability of exercise to prevent obesity resulting from excess caloric intake suggests that exercise may be valuable in the search for key energy balance regulators.

Subsequently, we compared the serum leptin concentration because leptin is involved in inhibiting food intake when food intake is regulated in the hypothalamus and is synthesized primarily in the white adipose tissue [4,14,19]. However, we found that greatly decreased leptin levels due to decreased body fat (data not shown) were not responsive to increased food intake as observed in exercise-trained mice fed an HFD. Physical activity may produce prolonged increases in leptin sensitivity and signaling in obese and diabetic rats through enhanced phospho-STAT3 expression and leptin receptor binding [20]. Thus, we concluded that exercise can increase leptin sensitivity in obese mice, which contributes to efficiently reducing food intake, even at low serum leptin levels. However, the opposite was indicated when high leptin levels in HFD control mice revealed increased food intake. Mice fed an HFD for 12 weeks appeared to have leptin resistance, in which leptin did not regulate food intake.

We found that the blood glucose concentration was significantly decreased in obese mice undergoing exercise training, although the insulin levels of these exercise-trained mice significantly decreased when compared with HFD-fed control mice. This finding is similar to the result from a report on diabetes mellitus [18] and the diabetic animal knockout model [9]. These research studies have reported that exercise increases muscle glucose uptake and glycogen synthase activity and directly reduces blood glucose levels by enhancing insulin signaling. This increases phosphatidylinositol 3-kinase (PI3-K), insulin receptor substrates (IRS), and glucose transport 4 (GLUT4) molecule expression. However, in healthy young men who underwent knee-extensor training for 3 weeks, their exercise training reduced IRS-1–associated PI3-K activity and did not generally improve insulin signaling afterward [21]. The apparent discrepancy between the effects of exercise is likely due to differences because of the frequency, strength, and length of exercise training, along with species variation. Altogether, exercise can modulate blood glucose levels and improve hyperglycemia, particularly in hyperglycemic humans and animals. Nevertheless, the effects of exercise on the response to insulin signaling in healthy animals that consume a low-energy diet may require investigation by using various short- and long-term exercise regimens.

Enhanced insulin sensitivity after exercise training was noted in obese animals fed an HFD, despite only having decreased insulin levels. Numerous studies have suggested that increasing adenosine monophosphate-activated protein kinase mediates the activity of surface GLUT4 translocation and is implicated in various antidiabetic and enhanced insulin sensitivity exercise properties [18,22]. These studies are consistent with our findings, which indicated that exercise reduces the insulin resistance index (assessed by HOMA-IR). In addition, exercise may reduce adiponectin from the adipose tissue inflammation factor, which is secreted from adipocytes and causes metabolic disturbances, including IR in obese mice [11]. The lower body weights of mice fed an HFD after exercise training for 12 weeks exhibited lower HOMA-IR indices. However, a low insulin concentration resulted in hypoglycemic mice fed an HFD undergoing exercise training, which greatly enhanced the β-cell function (assessed by the HOMA-β). This suggested that the HFD disturbed glucose homeostasis, which led to declining β-cell function and even β-cell atrophy, although exercise may have preserved or enhanced the β-cell function and mass through IRS 2 induction in diabetic rats [23]. Therefore, these results suggest that exercise can benefit IR and diabetic activity induced by an HFD.

Generally, exercise training can enhance fat oxidation associated with energy expenditure [24]. High exercise intensity significantly increases cardiac and skeletal muscle mass. In this study, the intensity and length of our exercise likely was a moderate–high-intensity continuous training regimen compared with the exercise training protocol used by Chavanelle et al. [25]. Moreover, daily and sustainable physical activity is feasible and effective for preventing excessive weight gain [12]. In contrast to sustainable exercise, short and unsustainable exercise do not greatly enhance energy expenditure, increase insulin sensitivity, and improve body fat [26]. Thus, regular and sustainable exercise may more efficiently induce numerous physiological benefits, facilitating decreased body fat accumulation and improved obesity, than short-term exercise does [27]. In this study, our animal model including a regular exercise protocol was meaningful to mimic a human model that applies exercise physiology, particularly with a sustainable endurance training regimen. However, this experimental model was limited in its long-term, high-frequency, and high-intensity exercise protocol.

Our results revealed that obese mice undergoing exercise exhibited a larger increase in the chromium concentration in the muscle, liver, and fat pads than did HFD-fed control mice, particularly in mice fed an HFD with high fasting glucose levels. However, chromium levels in the bone and blood indicated decreased fasting glucose levels. For essential metals, the bone is a physiological storage pool [4]. Chromium is released from bones and moved by the bloodstream into mobilizable pools, including the muscle, liver, and fatty tissues, which contribute to activating insulin signaling to reduce blood glucose levels [28]. Consistent with these observations, exercise promoted the movement of chromium from storage pools into glucose metabolic tissues. Reports have indicated that the skeletal muscle is a major site of insulin action [2,4], and exercise stimulates glucose uptake in the muscle, even in insulin receptor knockout mice [29]. This is consistent with mice undergoing exercise that exhibited a larger increase (*p* < 0.01) in the chromium concentration in the muscle than in other tissues in our study. Otherwise, exercise resulted in significantly increased hypoglycemia and increased plasma levels of lactate and β-hydroxybutyrate with a significant reduction of glycogen in the liver [30]. Exercise training improved glucose metabolism and was associated with reversing IR through decreased fat accumulation [31]. In addition, the blood glucose levels and serum insulin levels were not significant when mice were fed with an SD between exercise-trained and control mice (Appendix A). They indicated no significant differences in the chromium concentration in the blood and bones, muscle, liver, epididymal fat tissues, and kidney. In addition, the chromium intake of mice fed an HFD was lower than that of mice fed an SD (HFD 4.38 ± 0.06 versus SD 6.02 ± 0.05 μg/mouse/d; *p* < 0.01). To confirm whether the chromium levels were similar in the two groups fed an HFD or SD, we compared their chromium levels in the blood, bone, fat pads, muscle, liver, and kidney. The chromium levels did not present significant differences between mice fed an HFD and SD in the blood, bone, fat pads, liver, and kidney; however, significant differences were discovered between the two groups in the chromium level of the muscle (HFD 7.85 ± 0.20 versus SD 1.21 ± 0.05 ppb ×10^2^; *p* < 0.01). Chromium may enhance insulin sensitivity by increasing skeletal muscle cellular insulin signaling and improve the response of these alterations in the glucose metabolism, which contributes to overcoming HFD-induced hyperglycemia [4,7,32].

These mechanisms appeared through elevated chromium levels in glucose metabolic tissues linked to insulin action and subsequently facilitated the activation of insulin signal transduction to reduce blood glucose levels [10]. Thus, exercise may have caused the mobilization of chromium levels in tissues and organs, which prevented mice fed an HFD from developing symptoms of hyperglycemia in addition to acini–islet–acinar (AIA) axis communication, which is crucial for developing obesity and type 2 diabetes [33]. AIA feedback links the endocrine and exocrine parts of the pancreas and emphasizes the essential role a single organ plays in regulating glucose homeostasis by using amylase, most likely in the gut epithelium, and by using insulin and glucagon in peripheral blood [33,34]. Altogether, studies on chromium links to the AIA axis are necessary for assessing possible effects on obesity and diabetes treatment.

The kidney is the major organ that excretes essential trace metals [24]. In this study, HFD-fed mice that exercised exhibited higher chromium levels in the kidney than did control mice. This result is consistent with the finding of major reabsorption of trace metals in the renal proximal convoluted tubule [35]. However, another report indicated that exercise can increase chromium loss from urine during the normal physiological functioning of the body [36]. This apparent discrepancy indicated that urinary chromium excretion in response to exercise is related to the degree of physical fitness [37]. Overcoming hyperglycemia by enhancing chromium reabsorption in obese mice may cause chromium to move to the metabolic tissue and contribute to activating insulin signaling. Further research is required to solve this problem through analyzing urine samples in mice fed an HFD that underwent exercise.

Altogether, exercise resulted in the marked pathological mobilization of chromium from bone into circulation when an HFD caused lower chromium levels in metabolic tissues, whereas the chromium levels of metabolic tissues increased in parallel. A report suggested activating glucose transporter 4 trafficking and enhancing insulin-stimulated glucose transport [4,38]. Another report indicated that adding chromium picolinate and nicotinate supplementation (400 μg/d) with exercise training may help achieve significant weight loss and protection against coronary artery disease and noninsulin-dependent diabetes mellitus through modifying risk factors in young obese female mice [39]. However, chromium picolinate compound supplements (400 μg/d) did not affect exercise-induced changes in body weight, fasting plasma glucose, serum insulin, plasma glucagon, and cardiovascular health in obese women (aged 27–51 years) [11]. Chromium supplementation may enhance or not synergistically influence exercise-induced changes in metabolic effects that are dependent on the age and health of test participants. Studies with an expanded experimental design are necessary to assess whether combined chromium supplementation and exercise blunt or antagonize independent effects, as observed in animal models with differential metabolic conditions.

## 4. Materials and Methods

### 4.1. Animals and Diet

Male C57BL6/JNarl mice at 4 weeks of age were obtained from the Education Research Resource, National Laboratory Animal Center (NLAC), Taipei, Taiwan. These mice were grouped and fed a standard diet a high-fat diet (HFD, diet 592Z, 20.4% protein-enriched diet and modified Lab w/35.5% Lard, 1.12 ppm Cr, metabolizable energy 4.5 kcal/gm, PMI Nutrition International Inc., MO, USA) (Appendix A) for 12 weeks. Generally, mice were fed a high-fat diet for 4 weeks to obtain a diet-induced obesity model [28]. Mice were housed individually in microisolation cages on high-efficiency particulate air filtered ventilated racks (Rungshin IVC Systems, Taichung, Taiwan) with a controlled temperature (22 ± 1 °C), humidity (55 ± 5%), along with a 12 hr light/dark cycle. Mice had free access to drinking water and food for their maintenance. The body weights of all mice were measured at 12 weeks prior to being analyzed. All animal experiments were used in accordance with the guidelines of the Care and Use of Laboratory Animals, as recommended by the Taiwan government. The protocol for the experimental mice was also reviewed and approved by the Institutional Animal Care and Use Committee (IACUC) of the National Chung Hsing University (IACUC Approval No. 100-28). Animals were killed by an overdose of anaesthetic, urethane (1.2 mg/kg) combined with carbon dioxide.

### 4.2. Animal Experiment

From the age of 4 weeks, mice were fed an HFD and then further divided into two additional groups. One group received exercise protocol on an HFD, while the other was used as the control group. In addition, the initiate weights of the C57BL6/JNarl mice on a HFD did not show a significant difference between the exercise and control groups (17.09 ± 0.28 g for the exercise group versus 16.89 ± 0.23 g for the control on an HFD). For the exercise protocol, mice were run on a 10-lane motorized rodent treadmill (Sunpoint Co., Ltd., Taoyuan, Taiwan) and adapted to running at 20 m/min for 20 min per day [40] as they followed the 5 day/week training program while avoiding unexpected accidents. The mice could not be prompted to continue running through the use of moderate electric stimulation (less than 0.1 milliampere). After a 12-week exercise endurance training regimen, an analytical test that included blood glucose, tissue chromium, and insulin levels was performed as outlined below.

### 4.3. Measurement of Body Weight, Food, Energy, and Chromium Intake

The body weights and food intake of the mice were recorded and measured weekly. To estimate food consumption levels, food intake was assessed by weighing the food in each cage dispenser, including the food that had spilled on the floor of the cage. To estimate both the dietary energy intake and chromium intake, the result was combined with food consumption data. In addition, at the end of the study period, the animals were anesthetized while various tissues as well as serum were harvested for subsequent analysis.

### 4.4. Blood Glucose and Hormone Concentration Determination

After 12 weeks of feeding, blood samples were collected from the mice in their cages after 12 h of fasting. Throughout the aforementioned experiment, blood glucose was measured from the tail vein at the times indicated above using a One TouchTM glucose meter (LifeScan Inc., Milpitas, CA, USA). Serum leptin and insulin concentrations were measured by use of the mouse leptin ELISA kit and rat/mouse insulin ELISA kit (#90030 and #INSKR020; Crystal Chem Inc., Downers Grove, IL, USA).

### 4.5. Insulin Sensitivity and β-cell Function Indices

HOMAs analyze IR and β-cell function (HOMA-β) indices from insulin, and fasting glucose is widely used for estimating insulin resistance and the β-cell secretion function [4]. Moreover, the degree of insulin resistance is widely used for the baseline following the method described by HOMA-IR [41,42]. In particular, low HOMA-IR values indicate high insulin sensitivity, whereas high HOMA-IR values indicate low insulin sensitivity. Two fundamental features in the pathogenesis of type 2 diabetes are IR and pancreatic β-cell dysfunction, which involve failure in compensating for IR by secreting adequate insulin [43]. The evaluation of pancreatic β-cell function-related insulin secretion and the tissue insulin sensitivity modeling of fasting glucose and insulin pairs were used as an index of HOMA-β to calculate fasting insulin and glucose concentrations [44]. Therefore, we used the two indices to evaluate the IR and β-cell secretion function of mice after exercise training. HOMA-IR was calculated according to the following formula [45]: HOMA-IR = [fasting insulin (mU/L) × fasting glucose (mmol/L)]/22.5. Moreover, insulin secretion was calculated as the HOMA-β cell index according to the following equation: HOMA-β = [fasting insulin (mU/L) × 20]/[fasting glucose (mmol/L) − 3.5]. The models of both IR and insulin secretion were calculated based on the fasting values of plasma glucose and insulin, according to the HOMA method that was previously validated against clamp measurements.

### 4.6. Tissue Preparation and Chromium Concentration Analysis

After the experiment, all the blood and various tissue samples of the liver, gastrocnemius muscle, epididymal fat pad, and femur were isolated and stored at −20 °C until further analysis. After washing with saline, all the blood and tissue samples were blotted dry and weighed. Sample concentrations of chromium were determined as previously reported. The tissue samples were briefly digested in 65% nitric acid and heated at 100 °C for 1 h. Chromium levels were determined through graphite furnace atomic absorption spectrophotometry (Hitachi Z-2000 series polarized Zeeman atomic absorption spectrophotometer; Hitachi Co. Ltd., Tokyo, Japan). We measured each sample in triplicate, and the mean value was employed for analysis. Concentrations were expressed as μg/L (ppb) for each sample, which indicated the quantity of chromium in dry weight. In addition, the relative recovery rate of chromium was determined at 5 ppb of the limits of quantification levels by 94% (*n* = 5). The total levels of chromium in the samples were determined at R^2^ > 0.995 through a regression analysis of sample absorption data on a standard curve at concentrations of 1–500 ppb.

### 4.7. Statistical Analysis

Results are shown as mean ± SEM. The differences between the two groups were analyzed by a t-test for purpose of comparison. A P value of less than 0.05 was considered significant.

## 5. Conclusions

This study revealed that regular treadmill exercise may result in decreased body weight gain, decreased food and energy intake, daily food efficiency, and low leptin levels in mice fed an HFD. Collectively, exercise improved hyperglycemia, hyperinsulinemia, HOMA-IR, and HOMA-β indices in mice fed an HFD. Moreover, exercise enhanced insulin sensitivity and reduced IR, thus helping regulate glucose homeostasis. These benefits indicated that exercise effectively attenuated damage from HFD-induced obesity. In addition, exercise resulted in the marked mobilization of chromium from the bones and bloodstream into glucose metabolic sites, including the muscle, liver, and fat tissues. This movement of chromium may have enhanced insulin signaling and reduced obesity-induced hyperglycemia levels. Thus, exercise may be effective in preventing or lowering the possibility of obesity and reducing hyperglycemia associated with accelerated chromium redistribution.

## Figures and Tables

**Figure 1 molecules-25-01658-f001:**
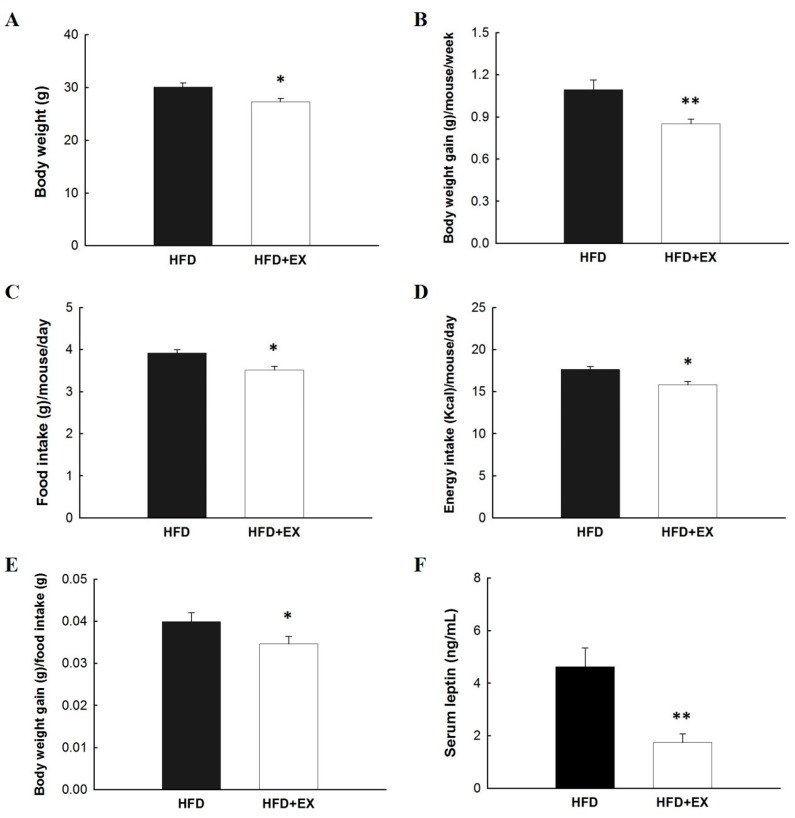
Effects of diet on (**A**) body weight, (**B**) body weight gain, (**C**) food intake per mouse day measured, (**D**) energy intake per mouse day measured, (**E**) daily food efficiency, and (**F**) serum leptin levels in mice after 12 weeks ad libitum consumption of high-fat diet (HFD). All values are given as mean ± SEM, *n* = 8 for all groups. Statistical significance at * *p* < 0.05 and highly statistically significant at ** *p* < 0.01.

**Figure 2 molecules-25-01658-f002:**
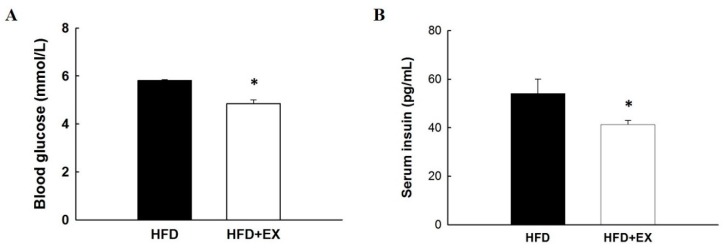
Effect of exercise in mice fed with a HFD on (**A**) blood glucose levels and (**B**) serum insulin levels. All values are given as mean ± SEM, *n* = 8 for all groups. Statistically significant at * *p* < 0.05.

**Figure 3 molecules-25-01658-f003:**
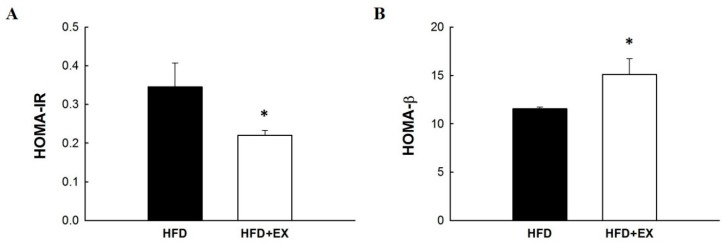
Effect of exercise in mice fed with an HFD on (**A**) homeostasis model assessment (HOMA)-insulin resistance (IR), and (**B**) HOMA-β indexes. All values are given as mean ± SEM, *n* = 8 for all groups. Statistically significant at * *p* < 0.05.

**Table 1 molecules-25-01658-t001:** Effects of exercise on levels of chromium in organs and tissues from control- or exercise-trained mice in a HFD.

Variable	HFD	HFD + EX
Blood (ppb) (×10^2^)	2.13 ± 0.10	1.88 ± 0.02 *
Bone (ppb) (×10^2^)	3.56 ± 0.26	2.79 ± 0.15 *
Muscle (ppb) (×10^2^)	7.85 ± 0.20	9.17 ± 0.33 **
Liver (ppb) (×10^2^)	1.21 ± 0.15	1.97 ± 0.28 *
Epididymal fat pads (ppb) (×10^2^)	1.07 ± 0.19	1.54 ± 0.13 *
Kidney (ppb) (×10^2^)	1.21 ± 0.06	1.57 ± 0.13 *

Data are presented as means ± SEM. Statistically significant at * *p* < 0.05 and highly significant at ** *p* < 0.01, compared with HFD control group. *n* = 8 for all groups.

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
