# Peer review of "Exercise Affects Blood Glucose Levels and Tissue Chromium Distribution in High-Fat Diet-Fed C57BL6 Mice"

_molecules, 2020, doi:10.3390/molecules25071658_

Round 1

Reviewer 1 Report

Dear Authors,

There was something to keep my attention and brink novel thinking when reading your manuscript “Exercise affects blood glucose levels and tissue chromium distribution in high fat diet-fed C57BL6 mice”.

On the first glance, however, I feel I am reading known facts.

Please explain in discussion:

  1. Does Cr blood/bone level in HFD mice was physiological? Show values of Cr of SD and SD + Exer mice. If the level of Cr was pathologically high in HFD mice, and exercise was a factor lowering it thus have primarily important factor which regulate IR.
  2. Yet another thinking was appeared when compare actual manuscript with quite aged paper of Solis- Heredia et al Toxicology, Chromium increases pancreatic metallothionein in rat. 2000, 3;142, (2), 111-7 whom observed that injected subcutaneously Cr salt increase in dose dependent manner blood glucose and alpha -amylase. The latest is of my special intertest when consider regulation of glucose – insulin – alpha amylase relation – see Pierzynowski et al . Glucose homeostasis dependency on acini-islet-acinar (AIA) axis communication: a new possible pathophysiological hypothesis regarding diabetes mellitus. Nutrition and Diabetes. 2018 Oct 8;8(1):55. As memento all diabetologist and gastroenterologist should recall that pancreas is single organ where both components hormonal and enzymatic affect each other very much. Without that consideration one can’t proceed with knowledge about obesity and diabetes.
  3. Minor point is – pls give to the reader short explanation what is HOMA-IR and HOMA-b.

Regards

Reviewer 2 Report

Comments to the Author

The manuscript entitled “Exercise affects blood glucose levels and tissue chromium distribution in high fat diet-fed C57BL6 mice” investigated the levels of blood glucose, plasma insulin, plasma leptin, food intake, body weight, tissue chromium concentration of mice fed a high-fat or a standard diet with or without exercise training. Although the study has demonstrated the chromium mobilization by exercise in high-fat diet-fed mice, there are serious concerns on data presentation and interpretation.

Major points:

  1. (Lines 33–35) “we suggest that exercise indeed prevents and controls HFD–induced obesity, and more effectively decreases fasting glucose levels through the regulation of chromium mobilization, which contributes to overcoming hyperglycemia.” This statement is not based on the presented data. This study only showed the association between chromium mobilization and improved glucose metabolism in mice fed the HFD. The authors did not investigate any causal relationship between them. The authors claim chromium mobilization from storage tissues into insulin target tissues. But we cannot know whether the mobilization plays a role in glucose metabolism or it is just a by-stander.
  2. (Table 1 and Supplementary Table 1) In SD-fed groups, exercise did not affect tissue chromium concentration. But some chromium concentrations in SD-fed mice are much different from those in HFD-fed mice. The data of standard diet-fed mice should be shown in the main figures and tables to compare with those of HFD-fed mice. The authors should discuss the effect of diet on chromium concentrations and glucose metabolism or insulin sensitivity.
  3. (Line 92) Compared with the HFD-fed mice without exercise, the HFD-fed mice with exercise showed a significantly decreased daily food intake, presumably leading to decreased chromium intake. I’m wondering whether the total (whole body) chromium content is changed by the diet or exercise.
  4. (Line 255) “Here, the exercised mice on an HFD showed an increase in kidney function relative to the control mice.” What does the data show us kidney function?

Minor

  1. (Line 86) “Surprisingly, body weight at the end of the treatment had been significantly reduced in the exercise-treated mice,” Why is this surprising? Exercise should reduce body weight gain.
  2. (Table 1 and Supplementary Table 1) HFD feeding in this study caused very mild obesity and glucose intolerance. The authors should confirm their findings in more severe obese mice.

Reviewer 3 Report

The manuscript presents very interesting topic. The study is well designed and described. I have only few issues: 1. Material and methods: what certified reference material has been used? and what accuracy has been for chromium? 2. In results and discussion it should be also mentioned that HFD diet influences on glucose and chromium status in mice. 3. Authors wrote that they used diet 592Z – citation should be provided; however in suppl. material Tab. 1 Authors wrote that diet 5008 has been used- it should be explained; it will be better to show the components of the diets in tables.

Round 2

Reviewer 2 Report

Major

  1. (Line 33) I would suggest that the sentence should be changed as “Thus, exercise prevents and controls HFD-induced obesity and may modulate chromium distribution in insulin target tissues.”

Minor

  1. (Line 246) (HFD 7.85 ± 0.20 vs SD 1.21 ± 0.05; ADD Units; P < 0.01);

Comment

Starting HFD at 4 weeks of age is too early, so that the mice don’t gain their weight much. I would recommend the feeding should be started after 7 weeks of age. Usually, the mice fed HFD for 12 weeks exhibit over 40 grams.
